# Can Corticomuscular Coherence Differentiate between REM Sleep Behavior Disorder with or without Parkinsonism?

**DOI:** 10.3390/jcm10235585

**Published:** 2021-11-27

**Authors:** Gyeong Seon Choi, Ji Young Yun, Sungeun Hwang, Song E. Kim, Jeong-Yeon Kim, Chang-Hwan Im, Hyang Woon Lee

**Affiliations:** 1Department of Neurology, Ewha Womans University School of Medicine, Seoul 07985, Korea; nselis@naver.com (G.S.C.); dream-yoon@hanmil.net (J.Y.Y.); neurosung@gmail.com (S.H.); 2Department of Neurology, Bundang Jesaeng General Hospital, Seongnam 13590, Korea; 3Department of Medical Science, Ewha Womans University School of Medicine, Seoul 07804, Korea; marielune@naver.com; 4Department of Biomedical Engineering, Hanyang University School of Engineering, Seoul 04763, Korea; swjy3565@naver.com (J.-Y.K.); ich@hanyang.ac.kr (C.-H.I.); 5Computational Medicine, Graduate Program in System Health Science & Engineering, Ewha Womans University, Seoul 03765, Korea

**Keywords:** polysomnography (PSG), electroencephalography (EEG), REM sleep behavior disorder, Parkinsonism

## Abstract

REM sleep behavior disorder (RBD) could be a predictor of Parkinsonism even before development of typical motor symptoms. This study aims to characterize clinical features and corticomuscular and corticocortical coherence (CMC and CCC, respectively) during sleep in RBD patients with or without Parkinsonism. We enrolled a total of 105 subjects, including 20 controls, 54 iRBD, and 31 RBD+P patients, patients who were diagnosed as idiopathic RBD (iRBD) and RBD with Parkinsonism (RBD+P) in our neurology department. We analyzed muscle atonia index (MAI) and CMC between EEG and chin/limb muscle electromyography (EMG) and CCC during different sleep stages. Although differences in the CMC of iRBD group were observed only during REM sleep, MAI differences between groups were noted during both REM and NREM N2 stage sleep. During REM sleep, CMC was higher and MAI was reduced in iRBD patients compared to controls (*p* = 0.001, *p* < 0.001, respectively). Interestingly, MAI was more reduced in RBD+P compared to iRBD patients. In comparison, CCC was higher in iRBD patients compared to controls whereas CCC was lower in RBD+P groups compared to control and iRBD groups in various frequency bands during both NREM N2 and REM sleep stages. Among them, increased CMC during REM sleep revealed correlation between clinical severities of RBD symptoms. Our findings indicate that MAI, CMC, and CCC showed distinctive features in iRBD and RBD+P patients compared to controls, suggesting potential usefulness to understand possible links between these diseases.

## 1. Introduction

The third edition of the International Classification of Sleep Disorders (ICSD-3) updated the diagnostic criteria for REM sleep behavior disorder (RBD), by repeated episodes of sleep-related vocalization and/or complex motor, and often violent, dream enactment behaviors, based on polysomnography (PSG) by demonstrating REM sleep without atonia (RWA) [1,2,3]. RBD shares clinical features with Parkinsonism, including Parkinson’s disease (PD), multiple system atrophy (MSA), and dementia with Lewy body (DLB) [4,5,6,7], which are often referred to as synucleinopathy. Moreover, a prospective study suggests that RBD is a preclinical marker of neurodegenerative disorders, especially in synucleinopathy [8]. Another study reveals evidence of neurodegeneration based on post-mortem pathology in RBD patients [9]. However, RBD is relatively rare in patients with tauopathy such as progressive supranuclear palsy (PSP) or Alzheimer’s disease (AD) [10]. Considering that PSP shares Parkinsonian motor symptoms with PD and AD shares commonalities in the existence of cortical dementia, it can be speculated that RBD is more specific to synucleinopathy rather than tauopathy.

Although idiopathic RBD (iRBD) patients do not show Parkinsonian motor symptoms, they sometimes present non-motor symptoms similar to PD [11,12]. PD patients display numerous non-motor features even before development of typical motor symptoms including neuropsychiatric signs and sleep disorders. The autonomic and sensory nervous and gastrointestinal systems can also display symptoms [12]. Idiopathic RBD (iRBD) patients also exhibit distinctive clinical symptoms, such as olfactory and/or autonomic dysfunction [11,12]. In a previous study, differences in motor measures, olfaction, and systolic blood pressure (BP) were found between iRBD and PD patients [13]. In this context, the severity of non-motor abnormalities could be intermediate between normal control and PD patients [13], and were suggested as biomarkers correlated with the severity of neurodegeneration [14,15]. In addition to non-motor clinical features, electrophysiologic findings could be used as markers to investigate the characteristics of iRBD and Parkinsonism. In a previous study using polysomnography (PSG), severity of REM sleep without atonia (RWA) predicted future development of PD [16,17]. In fact, the severity of RWA has been investigated using several methods—the muscle atonia index (MAI) is one of them, which has been updated for better quantification [18,19]. However, the specific clinical markers and/or specific criteria for phenoconversion from RBD to neurodegenerative diseases or precise mechanisms underlying the phenoconversion are not completely understood yet.

On the other hand, functional connectivity using various quantitative physiologic markers has been investigated to study physiologic and/or pathologic neural activities. Among various physiological markers, corticocortical coherence (CCC) and corticomuscular coherence (CMC) have been applied to investigate functional connectivity by measuring synchronized oscillatory activities between different cortical regions [20], or between cortical source to spinal motoneurons via the corticospinal tract [21,22]. More specifically, CCC quantifies the degree of synchronization between a pair of EEG signals recorded at different scalp locations, whereas CMC quantifies that between EEG and EMG signals, with the EEG signals being generally recorded above the motor cortex [23]. These measures could be helpful to investigate functional disruption in different cortical areas and/or corticospinal pathways in various neurological disorders. Disruption of interregional connections, for examples, was reported to correlate well with cognitive impairment in neurodegenerative diseases such as Alzheimer’s disease (AD), PD, and DLB [24,25,26]. In addition, impaired CCC and CMC have been suggested as electrophysiological hallmarks of PD [21], and can be modulated by treatment [24,26]. Recently, increased CMC during REM sleep has been proposed to be related to increased cortical locomotor drive in patients with RBD [21]. However, these previous literatures have been studied so far in neurodegenerative diseases, or only in RBD patients. Direct comparisons between iRBD and RBD plus neurodegenerative diseases such as PD or DLB have seldom been performed so far. In particular, no study has directly compared CMC and CCC between patients with iRBD only and those with both RBD and Parkinsonian features.

This study aimed to characterize the clinical and polysomnographic findings as well as electrophysiologic markers in RBD patients with or without Parkinsonism. To investigate the differences of corticocortical and corticomuscular connections including locomotor drive across stages of sleep, CCC and CMC, in addition to MAI were compared in control and RBD patients with or without Parkinsonism. We chose these indices because synchronization between different brain regions as well as corticomuscular connections could be involved as common pathophysiologic changes both in RBD and Parkinsonism patients, which would help us understand possible links between these diseases and underlying mechanisms of phenoconversion of iRBD to neurodegeneration.

## 2. Materials and Methods

### 2.1. Patients

All patients were recruited had visited outpatient sleep clinic at Ewha Womans University Mokdong hospital and been diagnosed with RBD between 2013 and 2019. Since RBD might be considered as one of the cardinal symptoms of neurodegeneration, especially for PD, we evaluated RBD patients on whether they had Parkinsonian symptoms or not. Specifically, we checked these patients for Parkinsonian motor symptoms such as bradykinesia, rigidity, tremor at rest, as well as postural instability, which was evaluated by a neurologist using Unified Parkinson’s disease scale part 3 (UPDRS-III) [27]. Sometimes, patients were referred from a movement disorder clinic who already had RBD symptoms when they were diagnosed as PD or DLB. PD and DLB were clinically diagnosed by an expert neurologist according to the diagnostic criteria of the United Kingdom PD Society Brain Bank [28] and the diagnostic criteria of the DLB Consortium [29].

Seventy-six RBD patients were identified. The diagnosis was based on the ICSD-3 criteria, which includes the existence of dream enacting motor activity associated with RWA, the presence of excessive tonic or phasic chin electromyographic (EMG) activity during REM sleep. Among them, 54 patients comprised the iRBD group and the remaining 22 were the RBD combined with Parkinsonism (RBD+P) group including 18 PD and 4 DLB patients. The disease duration was 6.9 years on average from the clinical diagnosis. As Control group, we recruited 20 age-matched healthy subjects who did not have a history of RBD or other chronic neuropsychiatric diseases.

We excluded subjects who had history of head trauma, encephalopathy, cerebrovascular disease, or severe diabetes mellitus; who were taking medication known to affect REM sleep including antidepressants and sedatives; or who had cognitive dysfunction with a Korean version of Mini-Mental Status (K-MMSE) score equal to or lower than 24; or who had respiratory disturbance index (RDI) higher than > 15/hour based on PSG, similar to the previous study [21]. Thus, a total of 96 subjects were finally included for further analyses, including 20 controls, 54 patients for iRBD, and 22 patients for RBD+P group.

### 2.2. Clinical Assessment

All evaluations were performed by a neurologist. All patients had completed the Korean version of REM sleep behavior disorder questionnaire (RBDQ-KR) for evaluation of clinical severity, based on previous literature [30,31,32,33], and the Beck Depression inventory (BDI), and had undergone an orthostatic blood pressure (orthostatic-BP) test. For the orthostatic-BP test, we checked the blood pressure in the supine position and after standing for 5 min. Systolic and diastolic BP decrease and heart rate variability were measured. Olfactory function was assessed using the Korean version of University of Pennsylvania Smell Identification Test (UPSIT) [34]. UPDRS-III [27,35] and K-MMSE were tested in all patients for both iRBD and RBD+P groups. The Hoehn and Yahr scale score was determined in the ‘on’ medication status by an expert neurologist in movement disorder clinic.

### 2.3. Sleep Studies

All participants attended a video-PSG recording using a comprehensive sleep laboratory with a Twin^®^PSG Clinical Software (Glass Technologies, Warwick, RI, USA). The video-PSG recording including 10–20 system electroencephalography (EEG), chin EMG, electrocardiography, snoring assessment, pulse oximetry, and airflow assessment was performed during one full-overnight monitoring in the sleep laboratory. Sleep stages were scored by a sleep specialist and experienced polysomnographers according to standard criteria of American Association Sleep Medicine (AASM) manual [36]. RBD was diagnosed on ICSD-3 criteria including the existence of dream enacting motor activity or PSG inspection of REM sleep-related behavior in association with the presence of excessive tonic or phasic chin EMG activity during REM sleep [1,2].

The PSG characteristics of RBD are characterized by either or both the following features: (1) Sustained muscle activity in REM sleep in the chin EMG or (2) excessive transient muscle activity during REM in the chin or limb EMG. Sustained muscle activity (tonic activity) in REM sleep was defined as an epoch of REM sleep with at least 50% of the duration of the epoch having a chin EMG amplitude greater than minimum amplitude than in NREM. Excessive transient muscle activity (phasic activity) in REM sleep defined when in a 30-s epoch of REM sleep divided into 10 sequential 3 s mini-epochs, at least 5 (50%) of the mini-epochs contain bursts of transient muscle activity. In RBD, excessive transient muscle activity bursts are 0.1–5.0 s in duration and at least 4 times as high in amplitude as the background EMG activity [26,30,34,37]. The analysis was based on a one-night PSG study and video analysis was performed at the same time for the assessment of exclusion of certain artifactual motor events.

### 2.4. Selection of PSG Segments from Relevant Sleep Stages

In previous literature, the atonia index and CMC were significantly different between control and iRBD groups only during REM sleep [21,33]. A more recent study showed that sleep spindles and sleep oscillations during NREM N2 and N3 revealed different features [23]. The reason we selected REM as well as NREM N2 was based on these studies. More specifically, we included NREM N2 to add REM-NREM comparison, but not N3, because N3 segments in our subjects were significantly decreased. In order to avoid sleep cycle bias, we calculated the indices from sleep segments during NREM N2 and REM sleeps from at least three different sleep cycles that were relatively early, mid-, and late PSG segments throughout the night, similar to the previous study [33]. Segments of NREM N2 and REM sleep stages were able to be found from three different sleep cycles consistently from all subjects, but not those from N3 sleep stage. Segments including visually detected movement or noise artifact and arousal were not analyzed. In order to avoid sleep cycle bias, REM sleep, in addition to N2 stage, representative for NREM sleep, were visually extracted in 30-s epochs adopted from at least three different sleep cycles from early, mid-, and late parts of full night sleep; early sleep cycle was defined as cycle 1, mid-sleep cycles as cycle 2–3 or 2–4, and late sleep cycles as cycle 4 or 5, as indicated previously [33,38].

### 2.5. EEG Recording and Preprocessing for Quantitative Analysis

In addition to the standard PSG, we recorded scalp EEG from 19 electrodes according to the international 10–20 system during the full-night sleep study. Similar to the previous studies [21,33], we used linked ear electrodes as a reference and the ground electrode placed on the forehead. EEG data were sampled at 256 Hz and bandpass filtered between 0.1 to 70 Hz, with a notch filter at 60 Hz and with impedances of all EEG electrodes less than 10 kΩ. EEG segments were selected only after visually detected movement or noise artifact and arousal were excluded, and NREM N2 and REM sleep segments were visually extracted in 30-s epochs adopted from at least three different sleep cycles to avoid sleep cycles bias, as described above.

### 2.6. Quantitative EMG/EEG Analysis: MAI, CMC, and CCC

We evaluated muscle atonia index (MAI), beta-band CMC between scalp EEG and chin/limb EMG, and CCC between intrahemispheric/interhemispheric electrodes. Ferri’s method was used to calculate MAI [18,19]. The EMG signal recorded from chin was rectified and averaged in 1-s mini-epochs. Then, we reduced noise/artifact in each mini-epoch. We used the minimum value found in a time window included 60 mini-epochs surrounding a current mini-epoch. We subtracted this value from the average rectified EMG amplitude of each mini-epoch [19]. The MAI was defined as the ratio between the EMG mini-epochs with corrected average amplitude ≤ 1 μV and the total EMG mini-epochs. Some epochs with 1 μV < corrected average amplitude ≤ 2 μV were excluded. The MAI ranged from 0 to 1. An atonia index value close to 0 indicates absence of EMG atonia, and a value close to 1 indicates stable EMG atonia. The MAI was calculated for N2 stage for NREM sleep and REM sleep.

In order to evaluate CMC, coherence was computed between four pairs of chin/limb EMG signals and central region EEG signals: chin EMG-C3 EEG, chin EMG-C4 EEG, right limb EMG-C3 EEG, and left limb EMG-C4 EEG. The Neurospec 2.0 Matlab toolbox (http://www.neurospec.org/) (accessed on 23 Mar 2019) was used to calculate the coherence after EMG and EEG signals were rectified in 30-s epochs to enhance the firing rate of the signals [22]. The coherence value was calculated for each epoch consisting of 6000 time samples. We averaged the results from the following equation applied to 23 non-overlapping sub-windows with 256 samples each (the remaining 122 time samples were discarded):$$C_{xy}\left(f\right)=\frac{{\left|P_{xy}\left(f\right)\right|}^{2}}{P_{xx}\left(f\right)P_{yy}\left(f\right)},$$

Here, $P_{xx}\left(f\right)$ and $P_{yy}\left(f\right)$ represent auto-spectra of *x* and *y*, respectively, and $P_{xy}\left(f\right)$ is the cross-spectrum of *x* and *y*. Coherence was calculated as the function of frequency f. The frequency range of interest was set to beta frequency band (from 12 to 30 Hz). Since the CMC values could be influenced by which muscles were measured [39], the representative CMC for each subject was presented as the average value of chin EMG-C3 EEG, chin EMG-C4 EEG, right limb EMG-C3 EEG, and left limb EMG-C4 EEG.

CCC was also calculated using the same procedure as described previously [33].

Coherence values between all pairs of EEG signals recorded from 19 electrodes were evaluated using the Neurospec 2.0 toolbox in five frequency bands based on conventional criteria [21,23,33]: delta (1–4 Hz), theta (4–8 Hz), alpha (8–12 Hz), beta (12–30 Hz), and gamma (30–55 Hz).

### 2.7. Standard Protocol Approvals, Registrations, and Patient Consents

The institutional Review Board (IRB) of Ewha Womans University School of Medicine approved this study. Requirement for informed consent was waived for this retrospective analysis of clinical data.

### 2.8. Statistical Analysis

Descriptive data are presented as mean ± standard deviation or frequency (percentage). Group comparisons of demographic, clinical, and PSG findings; CMC; MAI; and CCC among control, iRBD, and RBD+P groups were performed. Analysis of variance (ANOVA) or Kruskal–Wallis was used for evaluation of continuous variables and dichotomous variables, and post hoc significance was assessed with the Tukey–Kramer test using SPSS 20.0 (SPSS Inc., Chicago, IL, USA) with the statistical significance defined as *p* < 0.05.

## 3. Results

### 3.1. Clinical and PSG Characteristics

Overall, we enrolled 20 healthy controls, 54 patients for iRBD, and 22 patients for RBD+P group (18 PD and 4 DLB). Their clinical characteristics are summarized in Table 1. Systolic BP and olfactory function were more highly impaired in RBD+P patients than in either control or iRBD patients. Compared with iRBD patients, the UPDRS-III score was higher in RBD+P group. Systolic BP was more highly impaired in RBD+P patients than in iRBD patients or control subjects. Olfactory function was more impaired in RBD+P patients than the control or iRBD group patients.

In PSG findings, the numbers of patients with N3 stage NREM sleep in iRBD and RBD+P groups was reduced compared to that in the control group. PLM score was higher in both iRBD and RBD+P groups compared with the control group. The PSG findings are summarized in Table 2.

### 3.2. Corticomuscular Coherence (CMC) and Muscle Atonia Index (MAI)

The CMC was evaluated in each stage and is summarized in Table 3. During NREM2 stage, CMC values in RBD+P group were not significantly different from those in the control and iRBD groups (Figure 1A). In iRBD group, CMC values increased during REM sleep compared with those in control subjects. However, these REM sleep stage values were not different from those of the RBD+P group (Figure 1B). Interestingly, a high RBDQ score was correlated with a high CMC index during REM sleep in iRBD and RBD+P groups, indicating that CMC index represents RBD severity (Figure 1C,D).

The MAI was significantly decreased during both NREM2 and REM sleeps in the RBD+P group compared with iRBD group (Table 3). The iRBD group showed a lower MAI than controls during both NREM2 and REM stages, and the MAI decrease was greater in the RBD+P group than in the control and iRBD groups (Figure 1E,F). However, there was no relationship between RBDQ scores and MAI in either iRBD or RBD+P groups (Figure 1G,H).

### 3.3. Corticocortical Coherence (CCC)

The degree of coherence between intrahemispheric/interhemispheric electrodes is believed to indicate the power of the interconnections. Generally, it has been considered that theta and alpha bands reflect resting state activities during sleepiness and/or waking states, beta band for sensorimotor activities, whereas gamma band for higher cortical, often cognitive activities. During NREM N2 sleep stage, the iRBD group revealed a lower alpha (8–12 Hz) and gamma (30–55 Hz) band coherence in frontal, temporal, and parietal areas compared with control group, whereas higher alpha band coherence in frontal, central, and temporal areas, higher beta (13–30 Hz) band coherence in parietal and occipital areas, compared with RBD+P patients. In RBD+P patients, CCC in frontal, central, and temporal alpha power, and CCC in frontal, central, temporal gamma power during NREM2 sleep were lower than controls, and also CCC in frontal, central, and temporal alpha power, CCC in central, parietal, and occipital beta power, as well as CCC in frontal, central, temporal, and occipital gamma power were lower than the iRBD group (Figure 2A).

During REM sleep, iRBD patients showed a considerably higher CCC in theta and alpha bands in frontal, central, and occipital areas compared with control subjects. The RBD+P group showed a lower CCC in beta band in frontal and temporal areas, and lower CCC in gamma band in frontal, temporal, parietal, and occipital areas than the control group. In addition, the RBD+P patients had a lower delta (2–4 Hz), theta (4–7 Hz), alpha, beta, and gamma band CCC in central, temporal, parietal, and occipital areas compared with iRBD patients (Figure 2B).

## 4. Discussion

This study characterized clinical and polysomnographic findings in RBD patients with or without Parkinsonism. We also investigated the differences of muscular and cortical activities by determining MAI, CMC, and CCC during NREM N2 and REM sleeps in control and RBD patients with or without Parkinsonism. The main findings of this study were: (i) that the MAI was reduced in NREM N2 and REM sleeps in iRBD compared to controls, but more reduced in RBD+P compared to controls and iRBD groups. (ii) The CMC was significantly increased during REM sleep in iRBD compared to controls, and tended to be increased during REM sleep in RBD+P without statistical significance. (iii) CCC in theta and alpha frequency bands in frontal, central, and occipital areas were increased in iRBD compared to controls, and higher theta and alpha CCC in central, parietal, and temporal areas in iRBD compared to RBD+P group, and lower CCC in various frequency bands in RBD+P compared to controls and iRBD groups. Interestingly, higher CMC values during REM sleep were correlated with more severe RBD symptoms based on RBDQ-KR in both iRBD and RBD+P groups.

We observed clinical differences between iRBD and RBD+P groups in systolic BP drop and orthostatic abnormalities. Compared to the control group, the RBD+P group demonstrated significant olfactory dysfunction. A research group conducting a small case-control study previously reported on abnormalities of motor, olfaction, and autonomic function in iRBD patients [40].

PSG findings demonstrated that overall sleep quality did not show major differences among groups except for decreased amount of N3 sleep in iRBD and RBD+P groups. However, both iRBD and RBD+P groups showed higher PLM scores compared with the control group. The correlation between RBD and PLM has been identified in iRBD patients [41], and the correlation between PLM and Parkinson’s disease is well known [42]. There was a significantly greater number of PLMs in patients with PD and RBD compared to patients with PD but not RBD [43]. In PD patients, sleep-related movement disorders such as PLM and restless leg syndrome (RLS) also showed inverse correlations with disease duration and severity [44].

The CMC values showed significant differences between iRBD and control groups during REM sleep, but there were no differences among groups during NREM N2 sleep. The MAIs were distinct among groups during both NREM N2 and REM sleep stages, especially between iRBD and RBD+P groups. In previous studies, the iRBD group had higher beta frequency CMC and lower MAI than controls during REM sleep [21]. Beta-band oscillations between sensorimotor cortex and muscles implied locomotor control in human and monkey [45]. RBD patients with Parkinsonism present higher CMC in comparison to control during REM sleep. Our findings suggest that locomotor drive breaks down as the disease progresses. Interestingly, higher CMC values in beta frequency band during REM sleep were correlated with more severe RBD symptoms based on RBDQ-KR in both iRBD and RBD+P groups. These findings might add some benefits for future application of these indices to predict disease progression with further validations.

During REM sleep, the CCC values between iRBD and control groups documented significant differences in frontal and occipital areas. While differences between iRBD and RBD+P groups during NREM N2 sleep were recorded in faster frequency bands, differences between iRBD and RBD+P groups during REM sleep was prominent in slower frequency bands, mainly in motor and supplementary motor areas and in part of the premotor area. Previous study of EEG in RBD determined that RBD patients had considerable occipital frequency during REM sleep in spite of prominent beta power [46]. Concomitantly increased EEG alpha (8–10 Hz) and sigma (11–16 Hz) power were recorded during NREM N2 sleep in Parkinson’s disease patients [47]. The NREM N2 sleep EEG was the result of the intrinsic network of neurons and synaptic organization of corticothalamic circuits by regulatory systems [48]. A role for the sensorimotor cortex in shaping sensation perception via movement-generated mechanisms has been documented [49]. In particular, increased theta power was sensitive to early cognitive dysfunction [46].

To examine whether these indices could reflect clinical severity of the RBD symptoms, we examined the relationship between these values and scores of RBDQ-KR. Although the RBDQ is based on a subjective questionnaire, a score of RBD symptoms from patients and/or caregivers, it is commonly used as one of the standard diagnostic instrument [30], especially the Hong Kong version that was the most commonly used questionnaire [31]. The questionnaire reflects severity of RBD clinically by calculating mean scores of 10 RBD-related symptoms, which has been validated in many different languages worldwide including Korean version [32]. In addition, previous studies demonstrated that the clinical severity of RBD based on RBDQ was correlated between sigma band (12–15 Hz) desynchronization of scalp EEG in the central brain region in iRBD [32], or to severity of brain atrophy in RBD with neurodegenerative disorder [50]. A more objective index based on video analysis of PSG has been proposed in recent studies [51,52], which could be another way to examine the clinical severity more objectively that can be considered in the following studies.

The methodological issues should be also noted. We used rectified EEG signals to calculate CMC and CCC, based on the previous literature, where time-frequency analysis was also used to compare the differences between iRBD and control groups [33]. In the current study, coherence was computed mainly in the frequency domain, based on this. Although the rectification of the EMG signals is generally known to be an important step in the calculation of CMC, McClelland et al. argued that the process might be unnecessary and inappropriate in many circumstances [53]. Furthermore, Bigot el al. proposed a new statistical test to detect time-frequency correlations between EEG and EMG signals [54]. More improvement in methodologies should be considered in the future.

The sleep architecture of patients did not show major difference compared to controls, except significantly decreased NREM N3 and non-significant tendency of increased NREM N1 in iRBD and RBD+P groups compared to controls. Overall percentages of NREM N1 seem to be slightly increased and N3 to be decreased in our subjects compared to the reference data from previous literatures [55,56]. WASO tended to be relatively high in our patients for both iRBD and RBD+P groups. One possible explanation is the first night effect and another possibility is that some of our subjects might have physiological sleep changes due to old age or partly with common sleep disorder such as mild obstructive sleep apnea, because patients with iRBD and RBD+PD groups included old adults with average age over 60 years old. We recruited control subjects with similar age ranges and included subjects with less than 10/hour of RDI based on PSG if they did not have history of chronic sleep disorders or other neuropsychiatric disorders.

In summary, this study characterized clinical and polysomnographic characteristics of controls and patients with idiopathic RBD and RBD+P. We also investigated differences in locomotor driving and corticocortical coherence in controls, iRBD patients, and RBD+P patients. These findings showed clear distinctions between healthy control, iRBD, and RBD+P groups. The CMC values documented differences between controls and iRBD patients. In particular, the MAI and/or CCC values were more prominently different between iRBD and RBD+P groups, suggesting the possibility of development of neurodegenerative disorders in those with newly diagnosed RBD.

The limitations of this study may include the inability to detect developments or changes in the characteristics of the target population among the groups because the comparisons were performed in a cross-sectional study setting, not a longitudinal design. We only used subjective score for evaluation of RBD clinical severity, and the PSG data from only one sleep center. There is another possibility of resemblance of PD and other diagnoses such as atypical Parkinsonian disorders including PSP considering that it is difficult to differentiate PD from and PSP in early stages [57,58]. This might have some impacts on the results of RED+P group, considering that RBD is more prevalent in synocleinopathy (e.g. PD) than tauopathy (e.g., PSP). The RBDQ-KR scores tend to be rather low for iRBD and RBD+P groups in this study, which might influence the results to some extents since the severity of RBD symptoms should matter in this kind of analysis.

Although it was not a longitudinal study setting, the current study showed that CMC and MAI were different in iRBD and RBD+P groups compared to controls, which was somewhat confirmatory to the previous literatures. In addition, we found that the clinical severity of RBD revealed significant correlation with higher CMC values during REM sleep in both iRBD and RBD+P groups. However, CMC and RBDQ-KR are indirect and subjective parameters, and correlation between CMC and RBDQ-KR at cross-sectional time points could not directly apply to the prediction of the disease progression, for instance, the future development of PD in RBD patients. Although the MAI could be a more plausible index, the lack of correlation between MAI and RBDQ-KR in the current study suggests that further studies are needed to support its plausibility. Instead, we have added the point that some of CCC values for alpha and theta bands tend to be lower in iRBD+P compared with control and/or iRBD groups, suggesting that this index might be useful to evaluate the disease progression. However, it was a cross-sectional study design; it would not possible to conclude whether the findings indicated a disease progression or not.

Overall, future studies with longitudinal settings using these indices in larger sample sizes would be helpful to understand possible links between these diseases and to predict future development of degenerative diseases in iRBD patients.

Further studies in longitudinal settings using these indices would be helpful to understand possible links between these diseases and to predict future development of degenerative diseases in iRBD patients.

## 5. Conclusions

In conclusion, our findings indicate that CMC, CCC, and MAI based on PSG-EEG studies revealed distinct features in RBD with or without Parkinsonism. Furthermore, these indices may indicate a longitudinal biomarker useful to predict progression of Parkinsonism in iRBD patients with further validation based on the current observation of significant difference in iRBD and RBD-P groups.

## Figures and Tables

**Figure 1 jcm-10-05585-f001:**
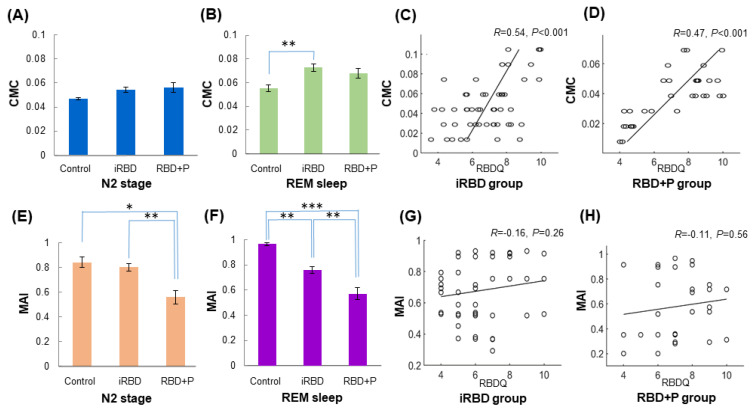
The corticomuscular coherence (CMC) and muscle atonia index (MAI) in control, iRBD, and RBD+P groups, with correlation between RBD severities. (**A**) CMC during NREM sleep, (**B**) CMC during REM sleep, (**C**) correlation between RBDQ scores and CMC values during REM sleep, (**D**) MAI during NREM sleep, (**E**) MAI during REM sleep, and (**F**) correlation between RBDSQ and MAI during REM sleep. Note that CMC values increased during REM sleep stage in iRBD group compared with control subjects, but was not different from those of the RBD+P group (**B**). Interestingly, the high score of RBDQ was correlated with high CMC index during REM sleep, both in iRBD and RBD+P groups, meaning that CMC index represents RBD severity (**C** and **D**). In addition, the MAI was significantly decreased during both NREM2 and REM sleeps in iRBD group, and even more reduced in the RBD+P group (**E**,**F**). However, there was no relationship between RBDQ scores and MAI in either iRBD or RBD+P groups (**G**,**H**). Abbreviation: CMC, corticomuscular coherence; MAI, muscle atonia index; iRBD, idiopathic RBD; RBD+P, RBD with Parkinsonism; NREM, non-rapid eye movement; REM, rapid eye movement; RBDSQ, RBD screening questionnaire; RBDQ, RBD questionnaire. Asterisks (*) means *p*-value < 0.05; ** *p* < 0.01, *** *p* < 0.001.

**Figure 2 jcm-10-05585-f002:**
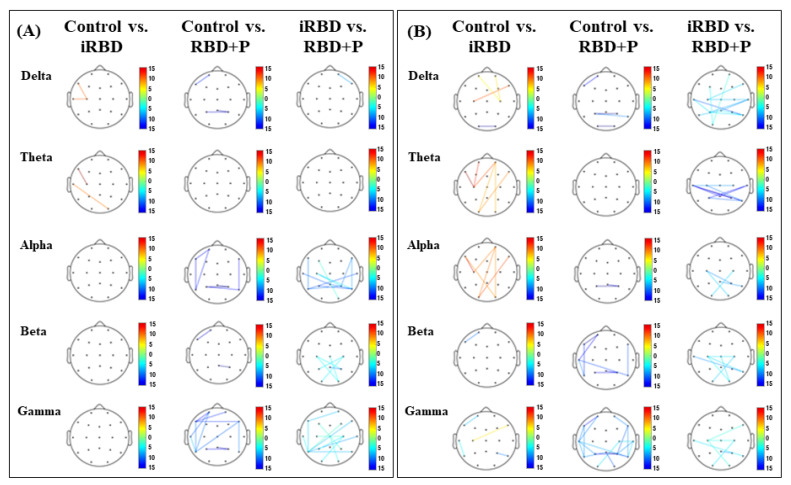
The CCC between electrode pairs shown with statistical significance at *p* < 0.05 during NREM N2 (**A**) and REM sleep (**B**). (**A**) During NREM N2 sleep stage, the iRBD group revealed a lower alpha and gamma band CCC in frontal, temporal, and parietal areas compared with control group, whereas higher alpha band coherence in frontal, central, and temporal areas, higher beta band CCC in parietal and occipital areas, compared with RBD+P patients. In RBD+P patients, CCC in frontal, central, and temporal alpha power, and CCC in frontal, central, temporal gamma power during NREM2 sleep were lower than controls, and also CCC in frontal, central, and temporal alpha power, CCC in central, parietal, and occipital beta power, as well as CCC in frontal, central, temporal, and occipital gamma power were lower than the iRBD group. (**B**) During REM sleep, iRBD patients showed a considerably higher CCC in theta and alpha bands in frontal, central, and occipital areas compared with control subjects. The RBD+P group showed a lower CCC in beta band in frontal and temporal areas, and lower CCC in gamma band in frontal, temporal, parietal, and occipital areas than the control group. In addition, the RBD+P patients had a lower delta, theta, alpha, beta, and gamma band CCC in central, temporal, parietal, and occipital areas compared with iRBD patients. Abbreviation: CCC, corticocortical coherence; NREM, non-rapid eye movement; REM, rapid eye movement; iRBD, idiopathic RBD; RBD+P, RBD with Parkinsonism.

**Table 1 jcm-10-05585-t001:** Demographic findings and clinical features in control, patient groups with idiopathic RBD and patients with combined RBD and Parkinsonism.

	Control (*n* = 20)	Idiopathic RBD (*n* = 54)	RBD with Parkinsonism (*n* = 22)	*p*-Value	Post-Hoc Significance
Age (yrs)	63.0 ± 8.0	62.2 ± 11.6	68.1 ± 9.5	0.080	
Sex (M:F)	11:9	32:22	12:10	0.091	
BMI (kg/m^2^)	23.9 ± 2.2	25.1 ± 4.0	23.1 ± 2.8	0.087	
RBDQ	0.6 ± 0.6	6.3 ± 2.4	7.0 ± 2.7	<0.001	* A < B * A < C
BDI	11.5 ± 9.1	12.5 ± 11.1	12.3 ± 10.1	0.962	
Motor function ^#^					
UPDRS-III			16.5 ± 10.2 (LEDD (mg/day) :372.8 ± 204.67)	(-)	
H&Y scale		0	2.0 ± 0.7	(-)	
Autonomic function					
Systolic BP drop (mmHg)	4.0 ± 2.0	11.5 ± 8.3	21.8 ± 14.4	0.010	* A < C * B < C
Diastolic BP drop (mmHg)	2.0 ± 1.1	5.3 ± 4.1	7.8 ± 7.8	0.416	
Cognition ^#^					
K-MMSE		28.8 ± 1.5	28.0 ± 1.8	0.253	
Special sense					
Olfaction (% normal)	55.6 ± 38.6	36.8 ± 31.6	13.6 ± 12.6	0.013	* A > C

Results are presented as mean standard deviation. RBDQ = REM sleep behavior disorder questionnaire; BDI = Beck Depression Inventory; UPDRS-III = Unified Parkinson’s Disease Rating Scale, part III; H&Y scale = Hoehn and Yahr scale for Parkinsonism; LEDD = Levodopa Equivalent Daily Dose; BP = blood pressure; K-MMSE = Korean version of Mini Mental State Examination; Olfaction was assessed with the brief University of Pennsylvania Smell Identification Test (UPSIT); ^#^ Evaluated in B and C groups; * *p* < 0.05.

**Table 2 jcm-10-05585-t002:** PSG findings in controls, patients with idiopathic RBD, and patients with combined RBD and Parkinsonism.

	Control (*n* = 20)	Idiopathic RBD (*n* = 54)	RBD with Parkinsonism (*n* = 22)	*p*-Value	Post-Hoc Significance
Total sleep time (min)	359.05 ± 79.52	342.9 ± 90.15	330.2 ± 90.7	0.359	
Time in bed (min)	459.73 ± 95.37	447.07 ± 99.71	462.48 ± 107.15	0.413	
Sleep latency (min)	23.0 ± 15.2	22.8 ± 17.0	23.8 ± 20.3	0.971	
REM latency min)	182.4 ± 107.5	140.2 ± 90.7	151.9 ± 90.1	0.238	
Sleep efficiency (%)	78.1 ± 14.2	76.7 ± 15.5	71.4 ± 18.3	0.334	
Arousal index (/hour)	15.6 ± 5.2	21.2 ± 16.3	21.1 ± 16.8	0.328	
REM sleep/total sleep time (%)	15.4 ± 6.33	15.5 ± 8.1	13.2 ± 7.3	0.192	
NREM sleep					
N1/total sleep time (%)	16.0 ± 13.1	23.3 ± 14.0	24.5 ± 16.6	0.111	
N2/total sleep time (%)	57.9 ± 11.9	52.9 ± 14.0	53.9 ± 19.1	0.436	
N3/total sleep time (%)	10.7 ± 8.02	8.2 ± 3.12	8.4 ± 4.1	0.007	* B < A * C < A
WASO (min)	43.7 ± 12.6	64.8 ± 16.3	72.7 ± 18.7	0.054	
RDI (/hour)	4.6 ± 4.5	12.3 ± 15.7	9.0 ± 9.2	0.071	
PLM score (/hour)	2.0 ± 3.8	4.6 ± 8.0	32.0 ± 46.6	<0.001	* B > A * C > A
PLM arousal (/hour)	0.4 ± 1.5	0.5 ± 1.9	2.8 ± 6.9	0.040	* B > A * C > A

Results are presented as mean standard deviation. * *p* < 0.05; REM: rapid eye movement; NREM: non-REM; WASO: wakefulness after sleep onset; RDI: Respiratory Disturbance Index; PLM: periodic leg movement.

**Table 3 jcm-10-05585-t003:** Corticomuscular coherence and atonia index in controls, patients with idiopathic RBD, and combined RBD with Parkinsonism.

	Control (n = 20)	Idiopathic RBD (n = 54)	RBD with Parkinsonism (n = 22)	*p*-Value	Post-Hoc Significance
CMC					
NREM N2 sleep	0.047 ± 0.036	0.054 ± 0.015	0.052 ± 0.009	0.154	
REM sleep	0.055 ± 0.011	0.073 ± 0.022	0.061 ± 0.012	0.002	** A < B
Atonia index					
NREM N2 sleep	0.842 ± 0.168	0.801 ± 0.205	0.621 ± 0.300	0.004	* C < A ** C < B
REM sleep	0.935 ± 0.390	0.761 ± 0.195	0.603 ± 0.262	< 0.001	** B < A ** C < B *** C < A

Results are presented as mean standard deviation. ** p* < 0.05, *** p* < 0.01, **** p* < 0.001; NREM: non-rapid eye movement; REM: rapid eye movement.

## Data Availability

All of the raw and processed data for clinical, polysomnographic, and quantitative electrophysiologic analysis were stored and can be accessed in our laboratory which supervised by the corresponding author (H.W.L.).

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
