# Peer review of "Can Corticomuscular Coherence Differentiate between REM Sleep Behavior Disorder with or without Parkinsonism?"

_jcm, 2021, doi:10.3390/jcm10235585_

Round 1

Reviewer 1 Report

Minor revisions

Line 58-- In fact, the seveity of 58 RWA has been investigated using several methods-- Spell check- revise to severity

Line 60-   the muscle atonia index (MAI) is one 59 of them, which has been updated for bettern quantification [17,18]. --- change to better quantification

Line 156-- In 155 order to avoid sleep cycle bias, REM sleep, in addition to N2 stage, representative for 156 NREM sleep, were visually extracted in 30-second epochs adopted from at least three dif- 157 ferent sleep cycles from early, mid- and late PSG segments; --- possibly say REM sleep fragments. Sentence otherwise sounds grammatically odd.

Line 210- Group comparions of dermographic, clinical and PSG findings;  -- Change to comparisons of demographic

Line 271 - The degree of coherence between intrahemispheric/interhemispheric electrodes believed to indicate the power of the interconnections.--- electrodes is believed

Line 332--- benifits for future application—spell check

Major revisions

  1. Line 66 - Among various physiological markers, corticocortical coherence (CCC) and corticomus- 66 cular coherence (CMC) have been applied --- It would be helpful to have the authors expand about CCC and CMC. It is not very clear from current description what these measures are, why and when they are used.
  2. Line 95- Please expand on how Parkinsonism was assessed by the authors based on chart review ? what measures were used ?
  3. Line 152 NREM N2 and REM sleeps from at least three different sleep cycles that were relatively 151 early, mid- and late PSG segments throughout the night, similar to the previous study [28]. --- Please clarify which previous study and expand on that here
  4. Line 220-- Compared with iRBD patients, the UPDRS-III score was higher in RBD+P 220 group-- What is the point the authors are trying to make here ? this is expected as one group has PD and the other does not. Goes back to the earlier point #2, how was Parkinsonism assessed?
  5. Line 366- Please clarify However, we noted that the rectification of the EEG signals has been argued in another 365 literature [49] and the importance of time-frequency depencence between non-stationary 366 signals for CMC and CCC [50], so improvement in methodologies should be considered 367 in the future.

  1. Please revise, the point you are trying to make here is not clear – Line 310 A large cohort study of RBD patients showed that abnormalities of motor, olfaction, and autonomic function were intermediated between controls 311 and patients with Parkinson’s disease

Author Response

Response to Reviewer 1 Comments

Point 1:

Minor revisions

Line 58-- In fact, the seveity of 58 RWA has been investigated using several methods-- Spell check- revise to severity

Line 60-   the muscle atonia index (MAI) is one 59 of them, which has been updated for bettern quantification [17,18]. --- change to better quantification

Line 156-- In 155 order to avoid sleep cycle bias, REM sleep, in addition to N2 stage, representative for 156 NREM sleep, were visually extracted in 30-second epochs adopted from at least three dif- 157 ferent sleep cycles from early, mid- and late PSG segments; --- possibly say REM sleep fragments. Sentence otherwise sounds grammatically odd.

Line 210- Group comparions of dermographic, clinical and PSG findings;  -- Change to comparisons of demographic

Line 271 - The degree of coherence between intrahemispheric/interhemispheric electrodes believed to indicate the power of the interconnections.--- electrodes is believed

Line 332--- benifits for future application—spell check

Response 1: We thank the reviewer’s time and efforts for those comments. All of the above have been corrected accordingly in the revised manuscript.

Major revisions

Point 2:

  1. Line 66 - Among various physiological markers, corticocortical coherence (CCC) and corticomus- 66 cular coherence (CMC) have been applied --- It would be helpful to have the authors expand about CCC and CMC. It is not very clear from current description what these measures are, why and when they are used.

Response 2: Thank you for your comments. The corticocortical coherence (CCC) measures oscillatory activities of cortical areas in different frequency bands, whereas the corticomuscular coherence (CMC) is regarded as efferent phenomenon, i.e., oscillations between the cortical sources to spinal motoneurons via the corticospinal tract. Therefore, CCC quantifies the degree of synchronization between a pair of EEG signals recorded at different scalp locations, whereas CMC quantifies that between EEG and EMG signals, with the EEG signals being generally recorded above the motor cortex. According to the reviewer’s suggestion, we have provided additional explanations about CCC and CMC in Introduction of the revised manuscript, on page 2, line 69~89, as follows:

“Among various physiological markers, corticocortical coherence (CCC) and corticomuscular coherence (CMC) have been applied to investigate functional connectivity by measuring synchronized oscillatory activities between different cortical regions [20], or between cortical source to spinal motoneurons via the corticospinal tract [21,22]. More specifically, CCC quantifies the degree of synchronization between a pair of EEG signals recorded at different scalp locations, whereas CMC quantifies that between EEG and EMG signals, with the EEG signals being generally recorded above the motor cortex [23]. These measures are known to help investigate functional disorganization in different cortical regions and/or corticospinal pathways in various neurological disorders. Disruption of interregional connections, for examples, was reported to correlate will with cognitive impairment in neurodegenerative diseases such as Alzheimer’s disease (AD), PD, and DLB [24-26]. In addition, impaired CCC and CMC have been suggested as electrophysiological hallmarks of PD [24], and can be modulated by treatment [25,26]. Recently, increased CMC during REM sleep has been proposed to be related to increased cortical locomotor drive in patients with RBD [21]. However, these previous literatures have been studied so far in neurodegenerative diseases, or only in RBD patients. Direct comparisons between iRBD and RBD plus neurodegenerative diseases such as PD or DLB have seldom been performed so far. Especially, no study has directly compared CMC and CCC between iRBD and RBD plus Parkinsonism patients.”

References:

[20] Strens, L.H.A.; Asselman, P.; Pogosyan, A.; Loukas, C.; Thompson, A.J.; Brown, P. Corticocortical coupling in chronic stroke: its relevance to recovery. Neurology 2004, 63, 475-484, doi: 10.1212/01.wnl.0000133010.69694.f8.

[21] Jung, K.Y.; Cho, J.H.; Ko, D.; Seok, H.Y.; Yoon, H.K.; Lee, H.J.; Kim, L.; Im, C.H. Increased Corticomuscular Coherence in Idiopathic REM Sleep Behavior Disorder. Frontiers in neurology 2012, 3, 60, doi:10.3389/fneur.2012.00060.

[22] Lieu, K.Y.; Sheng, Y.; Liu, H. Corticomuscular coherence and its applications: a review. Frontiers in Human Neuroscience 2019, 13, 100, doi.org/10.3389/fnhum.2019.00100.

[23] Myers, L.J.; Lowery, M.; O'Malley, M.; Vaughan, C.L.; Heneghan, C.; St Clair Gibson, A.; Harley, Y.X.; Sreenivasan, R. Rectification and non-linear pre-processing of EMG signals for cortico-muscular analysis. Journal of neuroscience methods 2003, 124, 157-165, doi:10.1016/s0165-0270(03)00004-9.

[24] Hirschmann, J.; Özkurt, T.E.; Butz, M.; Homburger, M.; Elben, S.; Hartmann, C.J.; Vesper, J.; Wojtecki, L.; Schnitzler, A. Differential modulation of STN-cortical and cortico-muscular coherence by movement and levodopa in Parkinson's disease. NeuroImage 2013, 68, 203-213, doi:10.1016/j.neuroimage.2012.11.036.

[25] Hammond, C.; Bergman, H.; Brown, P. Pathological synchronization in Parkinson's disease: networks, models and treatments. Trends in neurosciences 2007, 30, 357-364, doi:10.1016/j.tins.2007.05.004.

[26] Airaksinen, K.; Mäkelä, J.P.; Nurminen, J.; Luoma, J.; Taulu, S.; Ahonen, A.; Pekkonen, E. Cortico-muscular coherence in advanced Parkinson's disease with deep brain stimulation. Clinical neurophysiology : official journal of the International Federation of Clinical Neurophysiology 2015, 126, 748-755, doi:10.1016/j.clinph.2014.07.025.

Point 3:

  1. Line 95- Please expand on how Parkinsonism was assessed by the authors based on chart review? what measures were used?

Response 3: RBD is considered as one of prodromal PD features. Therefore, in our sleep clinic, we have been evaluating whether these patients with RBD symptoms have parkinsonian features or not. We have checked patients for parkinsonian motor symptoms such bradykinesia, rigidity, tremor at rest as well as postural instability. For more objective and semiquantitative measures, we have evaluated their UPDRS motor scales (MDSTF, 2003). For those patients who already were diagnosed as PD or DLB, diagnosis was made clinically by an expert neurologist according to the diagnostic criteria of the United Kingdom PD Society Brain Bank (Clarke et a., 2016) and the diagnostic criteria of the DLB Consortium (McKeith et al., 2017). All of these have been incorporated accordingly in Methods of the revised manuscript with relevant references, on page 3, line 100~110, as follows:

“We recruited patients who visited outpatient sleep clinic at Ewha Womans University Mokdong hospital and who were diagnosed as RBD between 2015 and 2019. Since RBD might be considered as one of the cardinal symptoms of neurodegeneration, especially for PD, we evaluated RBD patients whether they had Parkinsonian or not. Specifically, we have checked these patients for parkinsonian motor symptoms such as bradykinesia, rigidity, tremor at rest, as well as postural instability, which had been evaluated by a neurologist using Unified Parkinson’s disease scale part 3 (UPDRS-III)[27]. Sometimes, patients were referred from movement disorder clinic who already had RBD symptoms when they were diagnosed as PD or DLB. PD and DLB were clinically diagnosed by an expert neurologist according to the diagnostic criteria of the United Kingdom PD Society Brain Bank [28] and the diagnostic criteria of the DLB Consortium [30], respectively.

Reference:

[27] Movement Disorder Society Task Force on Rating Scales for Parkinson’s Disease. The Unified Parkinson’s Disease Rating Scale (UPDRS): Status and Recommendations. Movement Disorders 2003, 18, 738–750, doi:10.1002/mds.10473.

[28] Clarke, C.E.; Patel, S.; Ives, N.; Rick, C.E.; Woolley, R.; Wheatley, K.; Walker, M.F.; Zhu, S.; Kandiyali, R.; Yao, G.; et al. UK Parkinson’s Disease Society Brain Bank Diagnostic Criteria; NIHR Journals Library, 2016;

[29] McKeith, I.G.; Boeve, B.F.; Dickson, D.W.; Halliday, G.; Taylor, J.-P.; Weintraub, D.; Aarsland, D.; Galvin, J.; Attems, J.; Ballard, C.G.; et al. Diagnosis and Management of Dementia with Lewy Bodies: Fourth Consensus Report of the DLB Consortium. Neurology 2017, 89, 88–100, doi:10.1212/WNL.0000000000004058.

Point 4:

  1. Line 152 NREM N2 and REM sleeps from at least three different sleep cycles that were relatively 151 early, mid- and late PSG segments throughout the night, similar to the previous study [28]. --- Please clarify which previous study and expand on that here.

Response 4: Thank you for asking this important question. In order to avoid sleep cycle bias, we calculated CCC and CMC indices in PSG segments of relevant sleep stages from at least three different sleep cycles throughout the night, similar to one of previous studies by our colleagues (Sunwoo et al., 2019 & 2021). Early, mid- and late PSG segments meant that we selected sleep data from at least three different sleep cycles that were relatively early, mid-, and late parts of full night sleep; early sleep segment was defined as cycle 1, mid-sleep segments as cycle 2-3 or 2-4, and late sleep segments as cycle 4 or 5, after excluding visually-detected movement or noise artefact and arousals. To clarify the selection of sleep stages and segments, we have revised the description in great details in Methods under the section of “2.4. Selection of PSG Segments from Relevant Sleep Stages” on page 4, line 163~168 in the revised manuscript, as follows;

“In order to avoid sleep cycle bias, REM sleep, in addition to N2 stage representing NREM sleep, were visually extracted in 30-second epochs adopted from at least three different sleep cycles from early, mid- and late sleep cycles throughout the night for PSG study; early sleep cycle was defined as cycle 1, mid-sleep cycles as cycle 2-3 or 2-4, and late sleep cycles as cycle 4 or 5, as indicated previously [33,38].”

References:

[33] Sunwoo, J.S.; Cha, K.S.; Byun, J.I.; Kim, T.J.; Jun, J.S.; Lim, J.A.; Lee, S.T.; Jung, K.H.; Park, K.I.; Chu, K.; Kim, H.J.; Kim, M.; Lee, S.K.; Kim, K.H.; Schenck C.H.; Jung, K.Y. Abnormal activation of motor cortical network during phasic REM sleep in idiopathic REM sleep behavior disorder. Sleep 2019, 42, 1-10, doi:10.1093/sleep/zsy227.

[38] Sunwoo, J.S.; Cha, K.S.; Byun, J.I.; Jun, J.S.; Kim, T.J.; Shin, J.W.; Lee, S.T.; Jung, K.H.; Park, K.I.; Chu, K.; Kim, M.; Lee, S.K.; Kim, H.J.; Schenck C.H.; Jung, K.Y. Nonrapid eye movement sleep electroencephalographic oscillations in idiopathic REM sleep behavior disorder: a study of sleep spindles and slow oscillations. Sleep 2021, 1-10, doi:10.1093/sleep/zsaa160.

Point 5:

  1. Line 220-- Compared with iRBD patients, the UPDRS-III score was higher in RBD+P 220 group-- What is the point the authors are trying to make here ? this is expected as one group has PD and the other does not. Goes back to the earlier point #2, how was Parkinsonism assessed?

Response 5: Thank you for your comments. We absolutely agree on your opinion that higher UPDRS-III in RBD+P group is somewhat expected. We have removed this sentence in the revised manuscript as the reviewer suggested. As described earlier in Response 3, cardinal features for parkinsonian symptoms were evaluated by a neurologist based on UPDRS-III. PD and DLB were clinically diagnosed according to the diagnostic criteria of the United Kingdom PD Society Brain Bank and the diagnostic criteria of the DLB Consortium, respectively.

Point 6:

  1. Line 366- Please clarify However, we noted that the rectification of the EEG signals has been argued in another 365 literature [49] and the importance of time-frequency depencence between non-stationary 366 signals for CMC and CCC [50], so improvement in methodologies should be considered 367 in the future.

Response 6: Thank you for raising this issue. Although the rectification of EMG signals is generally known to be an important step in the calculation of CMC, McClelland et al. [53] argued that the process is unnecessary and inappropriate. Furthermore, Bigot et al. [54] proposed a new statistical test to detect time-frequency correlations between EEG and EMG data. We have revised the sentences accordingly on page 11, line 398~406, as follows:

“The methodological issues should be also noted. We used rectified EEG signals to calculate CMC and CCC, based on the previous literature, where time-frequency analysis was used to compare the differences between iRBD and control groups [33]. In the current study, coherence was computed mainly in the frequency domain, based on this. Although the rectification of EMG signals is generally known to be an important step in the calculation of CMC, McClelland et al. argued that the process in unnecessary and inappropriate in many circumstances [53]. Furthermore, Bigot et al. proposed a new statistical test to detect time-frequency correlations between EEG and EMG signals [54]. More improvement in methodologies should be considered in the future.”

References:

[33] Sunwoo, J.S.; Cha, K.S.; Byun, J.I.; Kim, T.J.; Jun, J.S.; Lim, J.A.; Lee, S.T.; Jung, K.H.; Park, K.I.; Chu, K.; Kim, H.J.; Kim, M.; Lee, S.K.; Kim, K.H.; Schenck C.H.; Jung, K.Y. Abnormal activation of motor cortical network during phasic REM sleep in idiopathic REM sleep behavior disorder. Sleep 2019, 42, 1-10, doi:10.1093/sleep/zsy227.

[53] McClelland, V.M.; Cvetkovic, Z.; Mills, K.R. Rectification of the EMG is an unnecessary and inappropriate step in the calculation of corticomuscular coherence. J Neurosci Methods 2012, 205(1), 190-201. doi: 10.1016/j.jneumeth.2011.11.001.

[54] Bigot, J.B.; Longcamp, M.; Maso, F.D.; Amarantini, D. A new statistical test based on the wavelet cross-spectrum to detect time-frequency dependence between non-stationary signals: application to the analysis of cortico-muscular interactions. Neuroimage 2011, 55(4), 1504-1518. doi: 10.1016/j.neuroimage.2011.01.033.

Point 7:

  1. Please revise, the point you are trying to make here is not clear – Line 310 A large cohort study of RBD patients showed that abnormalities of motor, olfaction, and autonomic function were intermediated between controls 311 and patients with Parkinson’s disease

Response 7: Thank you for your comments. The sentence you mentioned was unclear and redundant here, so we have removed it in the revised manuscript.

Again, we thank the reviewers and editors for their helpful comments and valuable suggestions. We feel that these revisions have significantly strengthened the manuscript, and hope that it is now acceptable for publication in the Journal of Clinical Medicine. Thank you very much for your kind consideration.

Sincerely yours,

Hyang Woon Lee, MD, PhD

Professor, Department of Neurology,

Director, Epilepsy and Sleep Center,

Ewha Womans University School of Medicine

Reviewer 2 Report

The study conducted by Choi et al aims to evaluate the differentiating potential of corticomuscular coherence (CMC) in the examination of RBD with and without parkinsonism. Authors revealed that in parkinsonian RBD muscle atonia index (MAI) was reduced when compared to iRBD. Corticocortical coherence was higher in iRBD when compared to controls, in RBD+P it was lower when compared to controls. The study showed that increased CMC during REM sleep is a feature revealing correlation between clinical severity of RBD symptoms. The main outcome of this study is the possible feasibility of MAI, CMC and CCC in the evaluation of iRBD and RBD+P. Some points should be addressed before further consideration:

  1. (Introduction) In the introduction authors elaborate on RBD in parkinsonian syndromes only in synucleinopathies. Though RBD shows up in synucleinopathies significantly more often than in tauopathies, a brief paragraph regarding the incidence of RBD in tauopathic parkinsonian syndromes should be added.

  1. (Patients) “the remaining 31 were the RBD combined with parkinsonism” – what were the parkinsonisms included in this study? How many patients had Parkinson’s Diseases (PD)? How many other diagnoses? What was the disease duration of the parkinsonism? What criteria of diagnoses were used?

  1. Perhaps a separate paragraph concerning limitations would be beneficial. Moreover apart from the limitations mentioned in the study, I would add the resemblance of PD and other diagnoses in certain stages as a limitation, as e.g. PD and PSP-P in the primary years are difficult to differentiate. This feature combined with the fact that PD is a synucleinopathy in which RBD is presents significantly more often than in PSP-P – tauopathy, should be stated as a feature possibly impacting results of RBD+P group. References, which in my opinion should be added:

  1. REM sleep behaviour disorder: how useful is it for the differential diagnosis of parkinsonism? doi: 10.1016/j.clineuro.2014.09.014. Epub 2014 Oct 5. PMID: 25459246.

  1. Progressive Supranuclear Palsy-Parkinsonism Predominant (PSP-P)-A Clinical Challenge at the Boundaries of PSP and Parkinson's Disease (PD). doi: 10.3389/fneur.2020.00180. PMID: 32218768; PMCID: PMC7078665

  1. Extensive language correction should be implemented

Author Response

Response to Reviewer 2 Comments

Point 1:

The study conducted by Choi et al aims to evaluate the differentiating potential of corticomuscular coherence (CMC) in the examination of RBD with and without parkinsonism. Authors revealed that in parkinsonian RBD muscle atonia index (MAI) was reduced when compared to iRBD. Corticocortical coherence was higher in iRBD when compared to controls, in RBD+P it was lower when compared to controls. The study showed that increased CMC during REM sleep is a feature revealing correlation between clinical severity of RBD symptoms. The main outcome of this study is the possible feasibility of MAI, CMC and CCC in the evaluation of iRBD and RBD+P. Some points should be addressed before further consideration:

  1. (Introduction) In the introduction authors elaborate on RBD in parkinsonian syndromes only in synucleinopathies. Though RBD shows up in synucleinopathies significantly more often than in tauopathies, a brief paragraph regarding the incidence of RBD in tauopathic parkinsonian syndromes should be added.

Response 1: Thank you for raising this important point. As the reviewer recommended, we have added previous literatures about RBD in tauopathy in the revised manuscript on pages 1~2, line 45~49, as follows:

“However, RBD is relatively rare in patients with tauopathy such as progressive supranuclear palsy (PSP) and Alzheimer’s disease (AD) [10]. Considering that PSP shares parkinsonian motor features and AD shares commonalities in the existence of cortical dementia, it can be speculated that RBD is more specific to synucleinopathy rather than tauopathy.”

Reference:

[10]      Nomura, T.; Inoue, Y.; Nakashima, K. Differences in Rapid Eye Movement Sleep Behavior Disorder Manifestation between Synucleinopathies and Tauopathies. Sleep and Biological Rhythms 2013, 11, 82–87, doi:10.1111/j.1479-8425.2012.00558.x.

Point 2:

  1. (Patients) “the remaining 31 were the RBD combined with parkinsonism” – what were the parkinsonisms included in this study? How many patients had Parkinson’s Diseases (PD)? How many other diagnoses? What was the disease duration of the parkinsonism? What criteria of diagnoses were used?

Response 2: Sorry for making a confusion. We enrolled 18 PD patients according to the United Kingdom PD Society Brain Bank clinical diagnostic criteria [5] and 4 probable or possible DLB patients according to fourth consensus report for DLB diagnostic criteria [6], giving a total number of RBD+P as 22. For those patients, the disease duration of parkinsonism was 6.9 years since the clinical diagnosis. Clinical diagnosis was made by an expert neurologist according to the diagnostic criteria of the United Kingdom PD society Brain Bank and the diagnostic criteria of the DLB Consortium, respectively. We have now corrected the number of patients and additional description about the diagnostic criteria in the revised manuscript to Methods section on page 3, line 101~111, and to Results section on page 3, line 115~119, as follows:

“All patients were recruited who visited outpatient sleep clinic at Ewha Womans University Mokdong hospital and who were diagnosed as RBD between 2015 and 2019. Since RBD might be considered as one of the cardinal symptoms of neurodegeneration, especially for PD, RBD patients were evaluated whether they had Parkinsonian or not. Specifically, we have checked these patients for parkinsonian motor symptoms such as bradykinesia, rigidity, tremor at rest, as well as postural instability, which had been evaluated by a neurologist using Unified Parkinson’s disease scale part 3 (UPDRS-III)[27]. Sometimes, patients were referred from movement disorder clinic who already had RBD symptoms when they were diagnosed as PD or DLB. PD and DLB were clinically diagnosed by an expert neurologist according to the diagnostic criteria of the United Kingdom PD Society Brain Bank [28] and the diagnostic criteria of the DLB Consortium [29], respectively.

“Among them, 54 patients comprised the iRBD group, and the remaining 22 were for the RBD combined with parkinsonism (RBD+P) group including 18 PD and 4 DLB patients. The disease duration was 6.9 years on average since the clinical diagnosis. As Control group, we recruited 20 age-matched healthy subjects who did not have a history of RBD or other chronic neuropsychiatric diseases.”

References:

[27] Movement Disorder Society Task Force on Rating Scales for Parkinson’s Disease. The Unified Parkinson’s Disease Rating Scale (UPDRS): Status and Recommendations. Movement Disorders 2003, 18, 738–750, doi:10.1002/mds.10473.

[28] Clarke, C.E.; Patel, S.; Ives, N.; Rick, C.E.; Woolley, R.; Wheatley, K.; Walker, M.F.; Zhu, S.; Kandiyali, R.; Yao, G.; et al. UK Parkinson’s Disease Society Brain Bank Diagnostic Criteria; NIHR Journals Library, 2016;

[29] McKeith, I.G.; Boeve, B.F.; Dickson, D.W.; Halliday, G.; Taylor, J.-P.; Weintraub, D.; Aarsland, D.; Galvin, J.; Attems, J.; Ballard, C.G.; et al. Diagnosis and Management of Dementia with Lewy Bodies: Fourth Consensus Report of the DLB Consortium. Neurology 2017, 89, 88–100, doi:10.1212/WNL.0000000000004058.

Point 3:

  1. Perhaps a separate paragraph concerning limitations would be beneficial. Moreover apart from the limitations mentioned in the study, I would add the resemblance of PD and other diagnoses in certain stages as a limitation, as e.g. PD and PSP-P in the primary years are difficult to differentiate. This feature combined with the fact that PD is a synucleinopathy in which RBD is presents significantly more often than in PSP-P – tauopathy, should be stated as a feature possibly impacting results of RBD+P group.

Response 3: Thanks for your important comments. Per the reviewer’s comments, we have edited a separate paragraph concerning possible limitations of the study, of which this has been added as another possible limitation in the revised manuscript on page 11, line 431~437, as follows:

“There is another possibility of resemblance of PD and other diagnosis such as atypical parkinsonian disorders including PSP considering that it is difficult to differentiate PD from and PSP in early stages [57,58]. This might have some impacts on the results of RED+P group, considering that RBD is more prevalent in synocleinopathy (d.g. PD) than tauopathy (e.g. PSP).”

References:

[57] Munhoz, R.P.; Teive, H.A. REM Sleep Behaviour Disorder: How Useful Is It for the Differential Diagnosis of Parkinsonism? Clin Neurol Neurosurg 2014, 127, 71–74, doi:10.1016/j.clineuro.2014.09.014.

[58] Alster, P.; Madetko, N.; Koziorowski, D.; Friedman, A. Progressive Supranuclear Palsy-Parkinsonism Predominant (PSP-P)-A Clinical Challenge at the Boundaries of PSP and Parkinson’s Disease (PD). Front Neurol 2020, 11, 180, doi:10.3389/fneur.2020.00180.

Point 4:

References, which in my opinion should be added:

  1. REM sleep behaviour disorder: how useful is it for the differential diagnosis of parkinsonism? doi: 10.1016/j.clineuro.2014.09.014. Epub 2014 Oct 5. PMID: 25459246.
  2. Progressive Supranuclear Palsy-Parkinsonism Predominant (PSP-P)-A Clinical Challenge at the Boundaries of PSP and Parkinson's Disease (PD). doi: 10.3389/fneur.2020.00180. PMID: 32218768; PMCID: PMC7078665

Response 4: Thanks for the useful information. References recommended by the reviewer have been added in the revised manuscript as indicated above in Response 3 (ref. #57 & #58).

Point 5:

  1. Extensive language correction should be implemented

Response 5: Thank you for your advice. We have asked for professional English proof reading provided by the JCM editorial office after the revision.

We thank the reviewer’s time and efforts for the comments. We have updated the revision with “Track Changes” function in the manuscript and provided the detailed response for each comment as indicated above. We feel that these revisions have significantly strengthened the manuscript, and hope that it is now acceptable for publication in the Journal of Clinical Medicine.

Thank you very much for your kind consideration.

Sincerely yours,

Hyang Woon Lee, MD, PhD

Professor, Department of Neurology,

Director, Epilepsy and Sleep Center,

Ewha Womans University School of Medicine

Reviewer 3 Report

The retrospective study investigated differences of neurophysiological features like MAI, CMC, CMS and PSG in patients with iRBD, RBD-P and controls. The patients also had UPSIT, BDI, MMSE, UPDRS III, RBDSQ, and orthostatic blood pressure assessment. 
Apparently patients just had one sleep study which should be mentioned in the methods chapter.
The methods should include video analysis for the assessment of exclusion of certain artefactual motor events. 
Table 1: The RBDSQ scores are rather low for iRBD and RBD-P, this should be commented, since the severity of symptoms may influence the results.
Table 2: N3 is too low for RBD and RBD-P patients and WASO is very high. This should be commented, since it differs from other RBD and RBD-P studies. Is this due to the conditions in the sleep lab or for other reasons? 
The results concerning MAI and their differences between groups in N2 and REM have been presented in many studies and are confirmatory. The same is true for UPDRS, UPSIT and orthostatic bp tests. The interpretation, that  correlation between CMC and RBDSQ-KR could predict disease progression should be reviewed. Both parameters are indirect or subjective parameters. Since this is a retrospective study and not a study with investigations at different time points this interpretation should be changed. The results of MAI are much more plausible and the lack of correlation with RBDSQ-KR underpin this questionable correlation mentioned aforehead. 
The coherence of alpha and theta band in frontal and temporal regions that are commented for iRBD  should also be presented for RBD-P patients in order to see if there is a progression, or if the changes are just random. This should also be done for the other bands, meaning that absolute values should be presented and not only group differences as trajectories.  The differences of the bands are given in the discussion and should be placed into results. The functional meaning of the band distribution (ie cognition) should be commented. 
Line 339: …had considerably occipital frequency …(which frequency?).
I think this study is confirmatory to a previous study and is important regarding the CCC and CMC findings. However, it has some problems concerning the symptom severity and the analysis of retrospective data. If authors should perform a future study a sample rate of 500-1000 should be performed to assess muscle activity in more detail. 

Author Response

Response to Reviewer 3 Comments

Point 1:

The retrospective study investigated differences of neurophysiological features like MAI, CMC, CMS and PSG in patients with iRBD, RBD-P and controls. The patients also had UPSIT, BDI, MMSE, UPDRS III, RBDSQ, and orthostatic blood pressure assessment. 
Apparently patients just had one sleep study which should be mentioned in the methods chapter.
The methods should include video analysis for the assessment of exclusion of certain artefactual motor events. 

Response 1: Thank you for your comments. Additional descriptions have been added in the revised manuscript accordingly, on page 3, line 141~143, and on page 4, line 158~160, as follows:

“The video-PSG recording including 10-20 system electroencephalography (EEG), chin EMG, electrocardiography, snoring assessment, pulse oximetry, and airflow assessment was performed during one full-overnight monitoring in the sleep laboratory.”

“The analysis was based on one-night PSG study, and video analysis was performed at the same time for the assessment of exclusion of certain artefactual motor events.”

Point 2:

Table 1: The RBDSQ scores are rather low for iRBD and RBD-P, this should be commented, since the severity of symptoms may influence the results.

Response 2: Per se the reviewer’s comments, it has been mentioned in the revised manuscript on page 11, line 435~437, as follows:

“The RBDQ-KR scores tend to be rather low for iRBD and RBD+P groups in this study, which might influence the results to some extents since the severity of RBD symptoms should matter in this kind of analysis.”

Point 3:

Table 2: N3 is too low for RBD and RBD-P patients and WASO is very high. This should be commented, since it differs from other RBD and RBD-P studies. Is this due to the conditions in the sleep lab or for other reasons? 

Response 3: Thanks for your comments. In the current study, percentages of each sleep stage are presented as values relative to total sleep, so they cannot be compared directly to those numbers that are calculated based on total sleep period, but they seem relatively comparable to the values in other reference data from European group [55]. WASO was presented as minutes, not %. We have corrected numbers and units, which has been incorporated in Table 2, on page 4~5 in the revised version. Now, the values of WASO in minutes seem similar to those values in the previous literature [56]. Relatively low percentages of N3 and high WASO might be related to the first night effects or some kind of selection bias because we recruited the control group with similar age ranges (over 60 years old) and excluded subjects with AHI scores higher than 10/min or with chronic neuropsychiatric disorders by history. We have mentioned this comment as the reviewer suggested in Discussion section of the revised manuscript on page 11, line 407-416 as follows;

“The sleep architecture of patients did not show major difference compared to controls, except significantly decreased NREM N3 and non-significant tendency of increased NREM N1 in iRBD and RBD+P groups compared to controls. Overall percentages of NREM N1 seem to be slightly increased and N3 to be decreased in our subjects compared to the reference data from previous literatures [49,50]. WASO tended to be relatively high in our patients for both iRBD and RBD+P groups. One possible explanation is the first night effect and another possibility is that some of our subjects might have physiological sleep changes due to old age or partly with common sleep disorder such as mild obstructive sleep apnea, because patients with iRBD and RBD+PD groups included old adults with average age over 60 years old.”

References:

[55] Hertenstein, E.; Gabryelska, A.; Spiegelhalder, K.; Nissen, C.; Johann, A.F.; Umarova, R.; Riemann, D.; Baglioni, C.; Feige, B. Reference data for polysomnography-measured and subjective sleep in healthy adults. J Clin Sleep Med 2018, 14(4),523-532.

[56] Ohayon, M.M.; Carskadon M.A.; Guilleminault, C. Meta-analysis of quantitative sleep parameters from childhood to old age in healthy individuals: Developing normative sleep values across the human lifespan. Sleep 2004, 27(7), 1255-1273.

Point 4:

The results concerning MAI and their differences between groups in N2 and REM have been presented in many studies and are confirmatory. The same is true for UPDRS, UPSIT and orthostatic bp tests. The interpretation, that correlation between CMC and RBDSQ-KR could predict disease progression should be reviewed. Both parameters are indirect or subjective parameters. Since this is a retrospective study and not a study with investigations at different time points this interpretation should be changed. The results of MAI are much more plausible and the lack of correlation with RBDSQ-KR underpin this questionable correlation mentioned aforehead. 

Response 4: Thank you for commenting this important issue. We agree on the reviewer’s opinion that correlation between CMC and RBDSQ-KR at cross-sectional time points could not directly apply to the prediction of the disease progression, for instance, the future development of PD in RBD patients. Future studies in longitudinal settings would be helpful to answer this kind of question. We have revised the sentences in the manuscript, on page 11, line 438~447, as follows:

“Although it was not a longitudinal study setting, the current study showed that CMC and MAI were different in iRBD and RBD+P groups compared to controls and that the clinical severity of RBD revealed significant correlation with higher CMC values during REM sleep in both iBRD and RBD+P groups, which are somewhat confirmatory of the previous literatures. However, CMC and RBD-SQ are indirect and subjective parameter, and correlation between CMC and RBD-SQ at cross-sectional time points could not directly apply to the prediction of the disease progression, for instance, the future development of PD in RBD patients. Although the MAI could be a more plausible index, the lack of correlation between MAI and RBD-SQ in the current study suggests that further studies are needed to support its plausibility.”

Point 5:

The coherence of alpha and theta band in frontal and temporal regions that are commented for iRBD should also be presented for RBD-P patients in order to see if there is a progression, or if the changes are just random. This should also be done for the other bands, meaning that absolute values should be presented and not only group differences as trajectories.  The differences of the bands are given in the discussion and should be placed into results. The functional meaning of the band distribution (ie cognition) should be commented. 

Response 5: Thank you for your comments. CCC was calculated in all electrode channel pairs in various frequency bands, so it was not easy to compare them between groups as for CMC and MAI to see the trends if there is a progression, for instance, from iRBD into RBD+P. Thus, we investigated the difference of each CCC value in corresponding electrode pairs for all different frequency bands between groups, respectively. That is why only CCC results with statistically significant differences between groups were presented as illustrated in Fig.2, e.g. group differences for corresponding electrode pairs in all different frequency bands. As the reviewer recommended, CCC differences of different frequency bands have been moved to Results under the section, 3.3. Corticocortical Coherence (CCC). Differences of CCC values in specific frequency tend to be lower in iRBD+P compared with control and/or iRBD groups, suggesting that this index might be useful to evaluate the disease progression. However, it was a cross-sectional study design, it would not be possible to conclude whether the findings indicated a disease progression or not, as the reviewer commented in Point 4. Please also note our answers in Response 4. We thought that CCC values for each electrode pair in various frequency bands are somewhat huge to be presented in a table in the main test, but we should be able to provide all the data in supplementary materials upon the reviewer’s request. Meanwhile, we have edited Results sections in the revised manuscript as the reviewer suggested, on page 8, line 295~314, as follows:

“The degree of coherence between intrahemispheric/interhemispheric electrodes is believed to indicate the power of the interconnections. Generally, it has been considered that theta and alpha bands reflect for resting state activities during sleepiness and/or waking states, beta band for sensorimotor activities, whereas gamma band for higher cortical, often cognitive activities. During NREM N2 sleep stage, the iRBD group revealed a lower alpha (8~12 Hz) and gamma (30~55 Hz) band coherence in frontal, temporal and parietal areas compared with control group, whereas higher alpha band coherence in frontal, central and temporal areas, higher beta (13~30 Hz) band coherence in parietal and occipital areas, compared with RBD+P patients. In RBD+P patients, CCC in frontal, central and temporal alpha power, and CCC in frontal, central, temporal gamma power during NREM2 sleep were lower than controls, and also CCC in frontal, central and temporal alpha power, CCC in central, parietal, and occipital beta power, as well as CCC in frontal, central, temporal and occipital gamma power were lower than the iRBD group (Fig.2A).

During REM sleep, iRBD patients showed a considerably higher CCC in theta and alpha bands in frontal, central and occipital areas compared with control subjects. The RBD+P group showed a lower CCC in beta band in frontal and temporal areas, and lower CCC in gamma band in frontal, temporal, parietal and occipital areas than the control group. In addition, the RBD+P patients had a lower delta (2~4 Hz), theta (4~7 Hz), alpha, beta and gamma band CCC in central, temporal, parietal and occipital areas compared with iRBD patients (Fig.2B).”

Point 6:

Line 339: …had considerably occipital frequency …(which frequency?).

Response 6: Thank you for asking this question. It was beta frequency range (from 12 to 30 Hz) in occipital ranges. We have added frequency information in the sentence on page 5, line 217~218, and on page 9, line 340~342, as follows:

“The frequency range of interest was set to beta frequency band (from 12 to 30 Hz).”

“Interestingly, higher CMC values in beta frequency band during REM sleep were correlated with more severe RBD symptoms based on RBDQ-KR in both iRBD and RBD+P groups.”

Point 7:

I think this study is confirmatory to a previous study and is important regarding the CCC and CMC findings. However, it has some problems concerning the symptom severity and the analysis of retrospective data. If authors should perform a future study a sample rate of 500-1000 should be performed to assess muscle activity in more detail. 
Response 8: Thank you so much for the reviewer’s comments. We have added this in Discussion as one of limitations of the current study and future direction on pages 11~12, line 438~454, as follows:

“Although it was not a longitudinal study setting, the current study showed that CMC and MAI were different in iRBD and RBD+P groups compared to controls, which was somewhat confirmatory to the previous literatures. In addition, we found that the clinical severity of RBD revealed significant correlation with higher CMC values during REM sleep in both iRBD and RBD+P groups. However, CMC and RBDQ-KR are indirect and subjective parameter, and correlation between CMC and RBDQ-KR at cross-sectional time points could not directly apply to the prediction of the disease progression, for instance, the future development of PD in RBD patients. Although the MAI could be a more plausible index, the lack of correlation between MAI and RBDQ-KR in the current study suggests that further studies are needed to support its plausibility. Instead, we have added this point that some of CCC values for alpha and theta bands tend to be lower in iRBD+P compared with control and/or iRBD groups, suggesting that this index might be useful to evaluate the disease progression. However, it was a cross-sectional study design, it would not possible to conclude whether the findings indicated a disease progression or not.

Overall, future studies with longitudinal settings using these indices in larger sample sizes would be helpful to understand possible links between these diseases and to predict future development of degenerative diseases in iRBD patients.”

Again, we would like to thank the reviewers and editors for their helpful comments and valuable suggestions. We carefully examined all the reviewers’ comments and have provided the detailed response point by point for each comment as indicated above. In addition, we have updated the revision with “Track Changes” function in the manuscript. We also have asked for professional English proof reading provided by the JCM editorial office after revision.

We feel that these revisions have significantly strengthened the manuscript, and hope that it is now acceptable for publication in the Journal of Clinical Medicine.

Thank you very much for your kind consideration.

Sincerely yours,

Hyang Woon Lee, MD, PhD

Professor, Department of Neurology,

Director, Epilepsy and Sleep Center,

Ewha Womans University School of Medicine

Round 2

Reviewer 2 Report

Authors have sufficiently addressed my concerns.

This manuscript is a resubmission of an earlier submission. The following is a list of the peer review reports and author responses from that submission.

Round 1

Reviewer 1 Report

Choi et al. present an investigation of muscular activity (measured with the muscle activity index, MAI), corticomuscular(CMC) and corticocortical (CCC) coherence in a cohort of iRBD, PD patients with RBD and healthy controls. The authors found: i) lower MAI in REM sleep in iRBD compared to controls and in RBD+P compared to iRBD and controls; ii) higher CMC in iRBD compared to controls; iii) higher theta and alpha CCC than controls in frontal, central and occipital areas and higher theta and alpha CCC in central, parietal and temporal areas compared to RBD+P. The study has a number of limitations, including methodological limitations. Such limitations make the quality of the study low. Below the main limitations are described:

  1. Introduction: The first paragraph consists only of the repetition of the same concept. The authors should just be clear and concise, by reporting the definition of the ICSD-3. It is surprising that the ICSD-3 is not cited.
  2. References: some articles in the introduction are not appropriately cited. For example, they cite the study [7] (Sixel-Doering et al., Sleep 2016) in the sentence “Prospective studies suggest that RBD is a preclinical marker of neurodegenerative disorders, especially in synucleinopathies…”. However, such study does not show evolution from iRBD to PD. In this context, the authors should cite appropriate studies, eg. Postuma et al., Brain 2019.
  3. Reference [15]: this is not the only study showing that RWA is a marker of phenoconversion. Other studies (eg. McCarter et al., Nuerology 2019) found the same.
  4. Sleep diagnoses: was the ICSD-3 applied? This should be specified. How was RWA quantified? Which cut-offs were used?
  5. The author state that the controls had “normal” sleep architecture. By looking at Table 1, the sleep architecture seems not normal. A mean WASO of 43.7% is reported for the controls and the mean N1% is over 20%. This is above normal data (see e.g. Ohayon et al., Sleep 2004). Furthermore, the authors do not specify whether these values are percentages of time in bed, total sleep time or sleep period time. The strange values of sleep stages make the reader wonder on the quality of the data.
  6. Probably the main issue of the manuscript consists of the fact that it seems that the author selected arbitrarily segments of N2 and REM sleep. This might have a strong influence on the results and constitute a bias. For example, despite removing epochs with movements, the authors might have selected the REM sleep epochs with more muscular activity in iRBD and RBD+PD (in fact muscular activity does not necessarily mean movement). An appropriate analysis should have been done by including all sleep epochs, removing only the segments (e.g. 3-s mini-epochs) with movement, noise artefact and arousal. Furthermore, for the selection of REM epochs, a bias might rise also from the selection of tonic/phasic REM (i.e. with or without rapid eye movement, see for example Sunwoo et al., Sleep 2019). Finally, how did the authors select “early, mid and late PSG segments”?
  7. The authors state that MAI was calculated for each sleep stage, but the results are not reported, except for the segments of N2 and REM sleep manually selected.
  8. It is not clear which are the results reported for CMC. The authors state that CMC was calculated for chin, left and right limb, but the results do not differentiate between these EMG channels. Furthermore, which limbs were used? Upper or lower limbs? It is known that upper limbs are more informative in RBD (Frauscher et al., Sleep 2012).
  9. Figure 1: the authors correlate CMC and MAI with RBDQ. Questionnaires are not objective, therefore the authors should consider this correlation with caution. If the authors want to correlate CMC with severity of RBD, such correlation should be performed based on the video analysis of the video recorded during vPSG (see for example Stefani et al., Sleep Med 2021).
  10. Figures 2 and 3: the colormaps are not readable. This is an important information that should be clear to the reader.
  11. Discussion: The first result that the author report is the difference in the PLMS index. It is pretty surprising that the authors state this as first result, as this is not the focus of the paper. The first paragraph of the discussion should report only the main results of the paper (i.e. MAI, CMC and CCC).
  12. In the abstract and conclusions, the author state that their findings “may help predict future development of Parkinsonism in iRBD patients”. This sentence should be deleted as this is not supported by the data, because the study is not longitudinal. Furthermore, significant differences between the three groups (which might indicate a longitudinal biomarker useful to predict progression) are found only for MAI (which is already well known), but not for CMC and CCC.
  13. In the presentation of the results and discussion, several times the authors talk about “NREM sleep”, but they actually refer only to the segments of N2 selected. This confuses the reader.
  14. An English check is also needed. For example in the Introduction: “In the previous study…” should be “In a previous study…”
  15. The authors do not stress what this study adds to the current literature on iRBD. This should be made much more clear to show the novelty of the study. 

Reviewer 2 Report

This study aims to find new biomarkers to diagnosis the development of Parkinsonism in people with idiopathic REM sleep behavior disorder (RBD). To this aim, the authors quantified several markers such as muscle atonia index (MAI), corticomuscular coherence (CMC) and cortico-cortical coherence (CCC) in RBD patients with or without Parkinsonism and healthy controls while they were sleeping. The results seem to indicate that biomarkers can distinguish groups of participants. According to the authors, these biomarkers may help predict future development of Parkinsonism in idiopathic RBD patients.

As a reviewer, I find that the manuscript is not clear enough in its present form and lacks critical information. In the introduction section, the presentation of the different biomarkers used later on in the study is too concise. Especially, authors need to develop the presentation of CMC and CCC. These markers have been the subjects of much debate (what does these markers represent, how they are computed, …) and its worth developing this part to better understand the rationale of using these. For instance, authors quantify CMC magnitude in the beta band only; just the choice of this frequency band may be arguable. It will also help formulating clearer hypothesis than they are currently formulated.

The method section lacks a lot of important information. Description of the system used to record EEG and EMG data are missing. EMG electrodes position is missing. How were identified the sleep stage? All the preprocessing of the EEG and EMG signals is missing. How much of the recordings have been discarded and for what purpose? How EEG signals were referenced? The rectification of the EEG signal is stated but is highly debated in the literature (see e.g., McClelland et al., 2012). I understand that coherence is computed in the frequency domain only; however, as EEG and EMG signals are non-stationary signals, time-frequency analysis should be preferred (see e.g. Bigot et al. 2011). How were chosen the frenqency bands of interest for the CCC analysis?

In the results section, a lot of or “clinical characteristics” results are presented but where not presented before, whether in the general problem, in the introduction section, or in the method section. These results are however discussed in the discussion section. I believe that the authors should bring more details regarding these clinical characteristics. Also, what is the rationale for focusing on the REM and NREM2 sleep period only? In the discussion section, results are presented again but they are not discussed. Why do RBD patients with Parkinsonism present higher CMC in comparison to control during REM sleep? How do the authors explained the CCC magnitude difference found in between the groups?

Finally, an intensive rewriting of the article will allow to better present the context and the variables measured and enhance the discussion of the results obtained. I recommend the authors to reduce the number of abbreviations to facilitate reading.